

# Incidence of ill-health related job loss and related social and occupational factors. The "unfit for the job" study: a one-year follow-up study of 51,132 workers

Francois-Xavier Lesage[1], Frederic Dutheil[2,3], Lode Godderis[4], Aymeric Divies[5] and Guillaume Choron[6]

[1] Epsylon, Univ Montpellier, Univ Paul Valéry Montpellier 3, CHU, Montpellier, France
[2] CNRS, LaPSCo, Physiological and Psychosocial Stress, University Hospital of Clermont-Ferrand, CHU Clermont-Ferrand, Preventive and Occupational Medicine, WittyFit, Université Clermont Auvergne, Clermont-Ferrand, France
[3] Faculty of Health, School of Exercise Science, Australian Catholic University, Melbourne, VIC, Australia
[4] Department of Public Health and Primary Care, Environment and Health, Katholieke Universiteit Leuven, Leuven, Belgium
[5] Occupational Health Service Ametra, Montpellier, France
[6] Univ Montpellier, CHU, Montpellier, France

Corresponding author
Francois-Xavier Lesage,
fx-lesage@chu-montpellier.fr

## ABSTRACT

**Objective:** The analysis of ill-health related job loss may be a relevant indicator for the prioritization of actions in the workplace or in the field of public health, as well as a target for health promotion. The aim of this study was to analyze the medical causes, the incidence, and the characteristics of employees medically unfit to do their job.

**Methods:** This one-year prospective study included all workers followed by occupational physicians in an occupational health service in the south of France. The incidence of unfitness for work have been grouped according to the main medical causes and analyzed. We performed a multivariate analysis in order to adjust the observed risk of job loss based on the age groups, sex, occupation and the activity sectors.

**Results:** A total of 17 occupational physicians followed up 51,132 workers. The all-cause incidence of being unfit to return to one's job was 7.8‰ ($n = 398$). The two main causes of being unfit for one's job were musculoskeletal disorders (47.2%, $n = 188$) and mental ill-health (38.4%, $n = 153$). Being over 50 years old (Odds ratio (OR) 2.63, confidence interval 95% CI [2.13–3.25]) and being a woman (OR 1.52, 95% CI [1.21–1.91]) were associated with the all-cause unfitness, independent of occupation and activity sector.

**Conclusions:** Identification of occupational and demographic determinants independently associated with ill-health related job loss may provide significant and cost-effective arguments for health promotion and job loss prevention.

## INTRODUCTION

The assessment of occupational diseases and their impact is a difficult issue. However, it is important for setting priority actions concerning health and safety at work. Most data concerning these occupational diseases indirectly assesses their incidence or impact. The findings of The Health and Occupation Reporting network (THOR) highlighted the gap between the epidemiological assessment by a network of specialists removed from the workplace (i.e., psychiatrists) and a direct assessment by an occupational physician linked to the workplace (*Carder et al., 2009*).

The difficulties with assessing the prevalence or the incidence of occupational diseases are (i) the comprehensive identification of the cases, (ii) the reliable notification of work causality concerning multifactorial diseases, and (iii) the knowledge about the population in whom these cases are identified. Over recent years, several occupational disease networks have been developed and data have been collected concerning occupational diseases. Several nationwide networks of clinical specialists have been developed in Europe (*Stocks et al., 2015*), such as THOR in the UK (*McDonald et al., 2006*), or the National Occupational Disease Surveillance and Prevention Network (RNV3P) in France (*Bonneterre et al., 2010*; *Faisandier et al., 2011*). Another way to assess occupational diseases is to match different data sources. *Mustard et al. (2015)* matched different and independent data sources: (i) emergency department encounter records, (ii) lost-time workers compensation claims, and (iii) representative samples of workers in a national health interview survey (*Mustard et al., 2015*).

Numerous countries may also have a compensation system for occupational diseases. Data concerning the recognition of occupational diseases are fairly easy to collect by the organizations which pay the compensation. These systems, however, may provide compensation to workers for benign illnesses, such as occupational contact eczema, but compensation may never be claimed for some frequent and fairly serious diseases, such as work-related depression or burnout syndrome. Consequently, this kind of indicator is a poor representation of the main occupational health problems in a country or a region. Likewise, the analyses of the causes of disability pensions or disability retirements are probably not very suitable for assessing the incidence of occupational diseases. As a result, two of the main difficulties are the collection and use of relevant and reliable indicators.

Unfitness ("inaptitude" in French), is the recognition by the occupational physician that a worker's health status is no longer fit for their current job and that they will require a job change. This does not mean that the worker cannot work anymore, but that they can no longer work in their current position. In France, the assessment of the fitness or the unfitness to perform the job is exclusively carried out by the occupational physician (OP) during a medical examination. All paid workers are systematically followed by the OP, either yearly or every two years. Additional specific medical examinations are carried out when a medical problem occurs. Moreover, when a paid worker is on sick leave for one month or more, fitness is systematically assessed by the OP when the worker returns to work (during a medical examination called

"the reinstatement visit"). An "unfit for the job" outcome may be pronounced for several reasons, including but not limited to: (i) a change in an employee's health status (e.g., returning to work after recovery from a serious illness or injury), (ii) a medical condition that may limit, reduce or prevent the person from effectively performing a new or current job (e.g., musculoskeletal conditions that limit mobility), (iii) a medical condition that is likely to make it unsafe both for the employee, their co-workers or the public (e.g., driving is essential to the job but the employee is subject to unpredictable and sudden unconsciousness). Each OP follows roughly 3,000 workers (*Dutheil et al., 2017*). There are currently 5,600 OPs in France who report one of three conditions back to the employer: fit, fit subject to work modification, or unfit for the job. Unfit for the job may lead to being assigned to a new suitable job in the same company, or to the termination of the employment contract due to medical issues. According to a recent study performed in one French region by the General Labor Department of the Ministry of Work, termination of the employment contract is the main conclusion of an unfitness notification (more than 97% of cases) (*Fernand, 2012*).

A previous study explored the analysis of the "unfit for the job" determination among 55,026 workers by collecting data directly from the medical records of a French occupational health service (*Dutheil et al., 2016*). The main limitation of this study was that only a univariate analysis was performed. Indeed, the data concerning the followed up workers were non-individual statistics and demographic information in aggregated form (e.g., sex ratio, occupations, average age), not a data frame extracted from the information system. Consequently, the socio-demographics could not be merged with the data frame of unfitness. However, to our knowledge, this is the only study analyzing ill-health related job loss.

The current study aimed to estimate the incidence of unfitness for the job in an occupational health service over a period of one year, and to describe their aetiology and the characteristics of the unfit employees.

## METHODS

### Study design and setting

This follow-up study took place between January 1st, 2014 and December 31st, 2014 in the occupational health service in Montpellier (France). Seventeen OPs followed up the employees of this employment area (except for the farming area and the employees in public services).

### Procedure and participants

All of the workers seen in this occupational health service in 2014 were eligible for participation in the study. General data concerning the workers were extracted from the information system: age, sex, length of service, occupation and industry, fitness or unfitness. Occupation was coded to four digits using the French Occupational Classification (PCS-2003 socio-professional categories) and the industries were coded using the European NACE-2008 nomenclature and aggregated A10 international classification. When an OP reported an "unfit for the job," they collected the age, sex,

socio-professional group, activity sector, and the diagnostic cause of the unfitness, which was supported by an expert medical opinion (e.g., by a psychiatrist, orthopedic specialist, or a rheumatologist depending on the disease) and the OP's own judgment on the work causality. The duration of the sick leave, the accident at work or the occupational disease recognition procedure, and what became of the employee (occupational reclassification, discharge due to unfitness) were also collected.

The subject's fitness for modified work or unfitness with occupational reclassification were not collected as an outcome. Each OP checked the cases of "unfit for the job" using an informatics query at the end of the year and completed the data collection if necessary. All data were anonymously collected. Participants were recruited during annual work medical examinations. No consent was given because anonymous data was used from normal daily clinical practices taken from medical records. The IRB approved the study (2018_IRB-MTP_02-02).

### Statistical analysis

The incidence of unfitness to work was calculated (numerator = number of unfit workers; denominator = number of followed up employees). The statistical analyses were conducted for all causes of unfitness and for the two main groups of pathologies. The relationship between unfitness and the other characteristics was calculated using the crude Odds Ratio and its 95% confidence interval (CI). Multivariate analyses were computerized, adjusting for age, sex, occupation and industry, using logistic regression models to provide the adjusted Odds Ratio (ORa). Statistical analyses were conducted using the R and epicalc packages (*R Core Team, 2015*; *Chongsuvivatwong, 2012*).

## RESULTS

Overall, 51,132 employees were followed up by 17 OPs in 2014 and were included in this study. Among them, 398 cases of "unfit for the job" were reported. The overall incidence of "unfit for the job" was 7.8‰.

Demographics and company characteristics of employees both fit and unfit for the job are presented in Table 1. The mean age of followed up workers was 38.7 years (SD = 11.8). The unfit for the job workers were older (mean = 44.4 years; SD = 11.8, $p < 0.001$).

When all causes were considered, (women ORa 1.52, 95% CI [1.21–1.91]) and workers over 50 years old (ORa 2.63, 95% CI [2.13–3.25]) were more frequently unfit for the job (Table 2).

Two groups of pathologies caused 85.7% of the cases of unfitness: musculoskeletal disorders (MSD) (47.2%) and mental ill-health (38.4%) (Table 2).

Most of the cases of unfitness were caused by a MSD. The average age of these workers was 46.0 years (SD = 11.3). Among the 188 cases of unfitness caused by a MSD, 32.4% were recognized as an occupational injury/disease, whereas 64.0% of these pathologies were estimated to be work related by the OP. The average length of sick leave was significant (13.1 months, SD 12.3). After adjustment, unfitness caused by a MSD was associated with gender (women ORa 1.89, 95% CI [1.35–2.66]) and the odds ratio increased with the age groups, notably that of over 50 year olds (Table 2). With regard

**Table 1 Demographics, company characteristics, and incidence of unfitness for the job.**

| | | Employees unfit for the job | Employees fit for the job | One-year incidence of unfitness ‰ |
|---|---|---|---|---|
| **Age (years)** | ≤30 year | 50 (12.6%) | 15,298 (30.2%) | 3.26 |
| | 31–40 year | 91 (22.9%) | 13,510 (26.7%) | 6.70 |
| | 41–50 year | 98 (24.6%) | 12,180 (24.0%) | 7.98 |
| | 51–60 year | 124 (31.2%) | 8,165 (16.1%) | 14.96 |
| | >60 year | 35 (8.8%) | 1,512 (3.0%) | 22.62 |
| **Sex** | Men | 124 (31.2%) | 21,279 (42.0%) | 5.79 |
| | Women | 274 (68.8%) | 29,455 (58.0%) | 9.22 |
| **Occupation** | Higher grade administrative and managerial occupations, higher grade professionals | 24 (6.1%) | 7,650 (17.0%) | 3.13 |
| | Intermediate occupations. Lower supervisors | 65 (16.5%) | 12,220 (27.2%) | 5.29 |
| | White collar workers. lower services, sales and clerical occupations | 212 (53.8%) | 19,671 (43.7%) | 10.66 |
| | Blue collar workers | 93 (23.6%) | 5,442 (12.1%) | 16.80 |
| **Workforce** | <10 | 105 (26.4%) | 13,000 (25.6%) | 8.01 |
| | 10–49 | 138 (34.7%) | 16,102 (31.7%) | 8.50 |
| | 50–249 | 93 (23.4%) | 14,503 (28.6%) | 6.37 |
| | ≥250 | 57 (14.3%) | 7,129 (14.0%) | 7.93 |
| **Activity sectors (aggregated A10 code)** | Manufacturing (BE) | 4 (1.0%) | 154 (0.3%) | 25.31 |
| | Construction (FZ) | 0 | 515 (1.0%) | 0 |
| | Wholesale and retail trade, repair of motor vehicles, transportation and storage, accommodation and food service activity (GI) | 123 (30.9%) | 14,051 (27.7%) | 8.68 |
| | Information & communication (JZ) | 2 (5.0%) | 2,306 (4.6%) | 0.87 |
| | Financial & insurance activities (KZ) | 9 (2.3%) | 2,954 (5.8%) | 3.04 |
| | Real estate activities (LZ) | 13 (3.3%) | 1,350 (2.7%) | 9.54 |
| | Professional, scientific and technical activities; administrative and support service activities (MN) | 69 (17.3%) | 9,696 (19.1%) | 7.07 |
| | Public administration, compulsory social security; education; human health and social work activities (OQ) | 152 (38.2%) | 14,909 (29.4%) | 10.09 |
| | Arts, entertainment, repair of household goods & other services (RU) | 26 (6.5%) | 4,799 (9.5%) | 5.39 |
| **Overall sample (number of workers)** | | 398 (100%) | 50,734 (100%) | 7.78 |

to the association with the industry, the activity sector for which the number of cases of unfitness was below 10 (manufacturing, construction, and information & communication) were excluded from the analysis. Moreover, the financial and insurance sectors were aggregated with real estate activities. This aggregated sector was considered as the reference

**Table 2 Factors associated with unfitness for the job, for the different causes of unfitness: univariate and multivariate analyses.**

| | | Causes of unfitness for the job | | | | | | | | |
|---|---|---|---|---|---|---|---|---|---|---|
| | | Musculoskeletal disorders (n = 188) | | | Psychopathologies (n = 153) | | | All causes together (n = 398) | | |
| | | ORc [CI 95%] | ORa [CI 95%] | p | ORc [CI 95%] | ORa [CI 95%] | p | ORc [CI 95%] | ORa [CI 95%] | p |
| Age (years) | ≤30 year | 1 | 1 | <0.001 | 1 | 1 | <0.01 | 1 | 1 | <0.001 |
| | 31–40 year | 1.76 [0.97; 3.25] | 2.37 [1.35; 4.18] | | 2.22 [1.34; 3.75] | 2.23 [1.36; 3.65] | | 2.05 [1.46; 2.90] | 2.43 [1.70; 3.45] | |
| | 41–50 year | 3.14 [1.83; 5.57] | 3.81 [2.25; 6.45] | | 1.96 [1.16; 3.38] | 1.90 [1.14; 3.18] | | 2.46 [1.74; 3.44] | 2.64 [1.86; 3.76] | |
| | 51–60 year | 6.56 [3.94; 11.39] | 7.35 [4.40; 12.28] | | 2.4 [1.38; 4.22] | 2.26 [1.32; 3.86] | | 4.59 [3.31; 6.37] | 4.75 [3.37; 6.69] | |
| | >60 year | 8.60 [4.22; 17.32] | 9.93 [5.06; 19.49] | | 3.24 [1.26; 7.42] | 3.22 [1.43; 7.23] | | 6.94 [4.52; 10.66] | 7.62 [4.85; 11.96] | |
| | ≤50 years | 1 | 1 | <0.001 | 1 | 1 | 0.07 | 1 | 1 | <0.001 |
| | >50 years | 3.65 [2.70; 4.91] | 3.41 [2.53; 4.61] | | 1.50 [1.02; 2.17] | 1.41 [0.98; 2.04] | | 2.82 [2.29; 3.46] | 2.63 [2.13; 3.25] | |
| Sex | Men | 1 | 1 | <0.001 | 1 | 1 | 0.14 | 1 | 1 | <0.001 |
| | Women | 1.64 [1.19; 2.25] | 1.89 [1.35; 2.66] | | 1.41 [1.00; 1.98] | 1.31 [0.92; 1.88] | | 1.48 [1.19; 1.83] | 1.52 [1.21; 1.91] | |
| Occupations | Intermediate occupations. Lower supervisors | 1 | 1 | <0.001 | 1 | 1 | 0.52 | 1 | 1 | <0.001 |
| | Higher grade administrative and managerial occupations, higher grade professionals | 0.11 [0.00; 0.69] | 0.12 [0.02; 0.88] | | 0.65 [0.34; 1.16] | 0.76 [0.43; 1.35] | | 0.59 [0.35; 0.96] | 0.66 [0.41; 1.06] | |
| | White collar workers. lower services, sales and clerical occupations | 4.47 [2.59; 8.27] | 3.88 [2.25; 6.70] | | 1.09 [0.74; 1.64] | 1.10 [0.74; 1.64] | | 2.03 [1.53; 2.72] | 1.95 [1.47; 2.60] | |
| | Blue collar workers | 9.43 [5.3; 17.85] | 8.13 [4.52; 14.62] | | 1.02 [0.56; 1.79] | 1.17 [0.66; 2.07] | | 3.21 [2.31; 4.49] | 3.23 [2.30; 4.54] | |

| | | Causes of unfitness for the job | | | | | | | | |
|---|---|---|---|---|---|---|---|---|---|---|
| | | Musculoskeletal disorders (n = 188) | | | Psychopathologies (n = 153) | | | All causes together (n = 398) | | |
| | | ORc [CI 95%] | ORa [CI 95%] | p | ORc [CI 95%] | ORa [CI 95%] | p | ORc [CI 95%] | ORa [CI 95%] | p |
| Activity sectors (aggregated A10 code) | Financial & insurance activities (KZ) | 1 | 1 | <0.01 | 1 | 1 | 0.25 | 1 | 1 | 0.20 |
| | Real estate activities (LZ) n = 4,326 | | | | | | | | | |
| | Wholesale and retail trade, repair of motor vehicles, transportation and storage, accommodation and food service activity (GI) n = 14,174 | **5.26** [2.00; 20.23] | **3.65** [1.32; 10.1] | | 1.01 [0.53; 2.06] | 1.05 [0.55; 2.01] | | **1.71** [1.08; 2.84] | 1.45 [0.91; 2.32] | |
| | Professional, scientific and technical activities; administrative and support service activities (MN) n = 9,765 | **4.22** [1.52; 16.28] | 2.36 [0.83; 6.71] | | 0.75 [0.36; 1.62] | 0.69 [0.34; 1.39] | | 1.39 [0.85; 2.37] | 1.01 [0.61; 1.65] | |
| | Public administration, compulsory social security; education; human health and social work activities (OQ) n = 15,061 | **4.33** [1.60; 16.42] | 2.33 [0.84; 6.44] | | 1.38 [0.75; 2.73] | 1.16 [0.63; 2.12] | | **1.99** [1.27; 3.28] | 1.35 [0.86; 2.13] | |
| | Arts, entertainment, repair of household goods & other services (RU) n = 4825 | 2.91 [0.90; 12.28] | 1.99 [0.64; 6.13] | | 0.76 [0.31; 1.84] | 0.76 [0.34; 1.70] | | 1.06 [0.58; 1.97] | 0.85 [0.48; 1.51] | |

**Notes:**
OR in bold = OR statistically different of 1.
ORc, crude Odds Ratio; ORa, odds ratio adjusted on age, sex, occupation and industry.

group. All of the other sectors were significantly associated with unfitness, but after adjustment, only one sector (trade, repair, transportation, accommodation and food services) remained statistically significant (ORa 3.65, 95% CI [1.32–10.1]) (Table 2). The group containing the higher grade white collar workers was less associated (ORa 0.12, 95% CI [0.02–0.88]) with unfitness than the reference group (intermediate occupations and lower supervisors). By contrast, the blue collar workers were highly concerned by unfitness resulting from a MSD (ORa 8.13, 95% CI [4.52–14.62]) compared with the same reference group (Table 2).

There were 153 declarations of unfitness caused by mental ill-health. Eight cases (5.2%) were recognized as an occupational injury or an occupational disease by the health insurance. Conversely, the OP estimated that 64.7% of these diseases were work related. Compared to MSD, the average length of sick leave (10.3 months SD 10.5) was lower ($p < 0.05$) and workers were younger (40.9 years versus 46.0, $p < 0.001$). Unlike for MSD, age and gender were less associated with mental ill-health related unfitness. Moreover, all occupation groups and activity sectors were concerned (Table 2).

## DISCUSSION

Data for this study were directly collected by the OP who determined whether the patient was "unfit for the job" or not. Consequently, the job losses and the medical causes are as reliable as possible. Moreover, they are based on a medical examination, supported by an expert medical opinion, and directly extracted from the medical records. Therefore, the data may be considered as reliable and of high quality.

The study design allowed information concerning the denominator to be completed. One strength of our design is that the population followed by the OP involved in this study is known and the requested information was extracted directly from the information system. Most of the experts in the network have accurate data concerning the cases (i.e., an occupational disease) but cannot collect reliable data on the population observed and, consequently, the indicators (e.g., incidence or associated risk).

Determining whether the patient's health is insufficient for the related job, known as "unfit for the job" in France, is exclusively performed by the worker's OP. The number of cases of ill-health related job loss is probably under-evaluated. For example, an employee with anxiety or depression caused by a conflict at work may choose to resign. He could also leave his job with the termination by mutual consent of employment contract. This employee would not be identified as "unfit for the job" although he should be recognized as "unfit for the job." A termination for a medical reason will maintain the unemployment insurance and termination indemnity. This is in the interest of a worker with a medical problem to be determined unfit for their job rather than to resign from the company. However, an employee do not always choose the most interesting way to leave his job in the specific context of mental ill-health caused by a conflict at work. Particularly severe or life-threatening diseases (e.g., most cancers) are probably under-evaluated in this study design, which is probably not appropriately adapted to reliably assess the impact of such pathologies on jobs.
Ill-health related job losses and socio-demographic data are easy to extract from the information systems using a simple SQL query. Most of the data are comprehensively coded in the databases of the occupational health services. Moreover, such an epidemiological system is very cost effective and is based on the French occupational health system.

One of the major advantages of integrating a denominator (the demographics of all the workers followed up by each OP) is the ability to carry out multivariate analyses. These analyses can provide risk assessments which would allow these risks to be monitored and geographical comparisons to be made in France, or other European countries. This would help to accurately identify ill-health related job losses.

The reader should be aware that this study design does not identify work related diseases, neither can it provide a job loads attributable risk, but rather assesses the impact of diseases, work-related or not, on the capacity to maintain the current job.

In our opinion, this must not be considered as a weakness of this study. The mechanisms of the observed pathologies may be partly, completely or not at all work-related. However, it is a delicate issue to determine if a disease is work-related or not (the work causality) for each individual. Diseases are often multifactorial with personal, professional and extra professional components (e.g., mental ill-health). In fact, all data concerning the issue of work causality should be cautiously interpreted. In our study, the outcome is job loss, which is an objective data, rather than the work causality of cases, which requires a judgement. The analysis of the "unfitness for the job," even if it does not allow the work related causality to be determined, allows at-risk groups to be identified and preventative actions to be promoted in order to support the continuation of employment.

### Findings

The global one-year incidence of unfitness in our study was 7.8‰. A similar study of ill-health related job losses was previously conducted in 2012 in another occupational health service in the east of France (*Dutheil et al., 2016*). Dutheil et al. found very similar results. They found that the overall one year incidence of job losses was 7.7‰. Dutheil's study confirmed the major impact of MSD and mental ill-health on employment. However, the design of this previous study did not allow multivariate analysis (c.f., above). Our study design allowed multivariate analysis because there was a single data frame for fit and unfit workers, and consequently the possibility to adjust the Odds Ratios based on the age groups, sex, occupation and activity sector, which are strongly associated with job losses. Our findings highlight that 85.7% of ill-health related job losses are related to MSD (47.2%) and mental ill-health (38.4%) (Table 2). The high prevalence of job losses related to MSD (*Lederer, Weltle & Weber, 2001*; *Cherry et al., 2001*; *Chen, McDonald & Cherry, 2006*) and mental ill-health (*Lederer, Weltle & Weber, 2001*; *Cherry, Chen & McDonald, 2006*; *Reinhardt, Wahrendorf & Siegrist, 2013*) have been estimated in other countries. Recent data from the THOR-GP network (UK—2015-2016) estimated that 85% of self-reported work related ill-health is caused by these two groups of pathologies (*Carder et al., 2013*), which is similar to our findings. This supports the relevance of

"unfit for the job" as an indicator, even if it does not assess the work related incidence of different pathologies. Moreover, the process of rating a pathology as work related is partly subjective and complex, and may lead to a misclassification.

The Labour Force Survey (LFS) is the Health and Safety Executive's data source, complemented by other sources such as death certificates and reports from doctors (THOR). It estimated the incidences of work related ill health by stress-anxiety-depression and MSD as 690/100,000 workers and 550/100,000 respectively (*Health and Safety Executive, 2017*).

The incidences were higher than the job loss incidence because all of these cases of work-related ill health did not lead to a classification of "unfit for the job." Stress, anxiety, and depression accounted for 37% of all work related ill health cases in 2015/2016, and MSD accounted for 41% of all work related illnesses. These data suggest that mental ill-health and MSD are not only the two main causes of work related diseases, but are also the two main causes of job loss.

The underlying employment area could not be representative of France as a whole. That is why we didn't extrapolated our findings, even if a previous study (*Dutheil et al., 2016*), conducted in another region of French, found a very similar incidence of job loss. Moreover, farming and public service employees (and to a certain extent the construction industry) are followed up by specific health services. Further study is needed to assess the ill health related job losses in these specific activity sectors. However, it would be interesting to extrapolate the data based on socio-professional categories to assess the burden of ill health related job loss in France.

## Musculoskeletal disorders

Our findings highlight the significant increase in the risk of job loss caused by musculoskeletal disorders in workers over 50 years of age. This is a major risk among blue collar workers. It is a particularly worrying situation for people over the age of 50 because the capacity to find a new job in activity sectors where mechanical loads are frequent, and one which is adapted to the potential disability caused by the MSD, may lead to employment problems for the concerned workers and significant social problems. Unfortunately, in our opinion, primary prevention for workplace tasks is probably rarely or insufficiently adapted to the age of the worker. However, maintaining working ability in early old age is essential for sustaining economic growth in Europe (*Reinhardt, Wahrendorf & Siegrist, 2013*).

## Mental ill-health

Our findings highlight the huge impact of mental ill-health on job loss. These diseases are invisible in the national data on occupational diseases (e.g., the recognized occupational diseases), notably because it is probably very difficult to determine work causality. For example, according Wong et al., attribution of mental illness to work is thought to involve the consideration of 18 factors (11 workplace factors and seven personal vulnerability factors) (*Wong, Poole & Agius, 2015*). In this study, work causality is determined by the OP's own judgement and their knowledge of the workplace. According to French medical

insurance data, only a few hundred cases of anxiety or depression are recognized as an occupational disease by the medical insurance.

Another important finding is that all workers are at risk. There may be an increased risk for women or older workers, but all socio-professional statuses and activity sectors are impacted by this issue.

## Sex

The results highlight the gender inequalities in the face of job loss due to ill-health (Table 2).

Dutheil et al. (2016) observed similar univariate risks for job loss due to MSD, mental ill-health and all-causes together for female workers (RR = 1.51, 1.70 and 1.51 respectively).

This gender inequality issue probably needs particular attention. One reason for such inequality may be that a high proportion of women work in activity sectors with high physical or psychological loads. For example, according the National Statistics Institute, women constitute 87% of nurses and 62% of unskilled workers. There are too few cases of job losses in this study (398 cases) to analyze the risks associated with specific occupations matched with sex ratio. However, an extended study (only 17 OPs participated in this study versus over 5,000 OPs in France) could provide more accurate information. A more specific analysis of activity sectors and jobs for the observed population could provide more accurate "indications" concerning these observed data.

## Age

Our findings concerning the average age of unfit workers (44.4 years), and the risks of unfitness among workers aged 50 years and older according to the disease groups (Table 2), are similar to the findings of the previous study carried out in 2012 (Dutheil et al., 2016). In this study, the average age of unfit workers was 45.9 years, and the risks of job loss for workers up to the age of 50 resulting from a MSD, a mental ill-health and all causes together, were 2.92, 1.38, and 2.51 respectively. Not surprisingly, our findings highlight the progressive increase of the odds ratios of job loss with the age of the workers, particularly for MSD.

The findings by Alavinia et al., which highlighted the higher relative risk of long term absence due to sickness among those $\geq$ 50 years old (RR = 2.08 [1.33;3.24]; ref = workers < 40 years old), are consistent with our data (Alavinia et al., 2009). Older workers are a more fragile population, with a higher prevalence of chronic and disabling pathologies leading to long term absence due to sickness and a higher incidence of job loss. Moreover, the combination of a disabling pathology and the loss of a job is a problematic cocktail for professional reintegration. In our opinion, this population is a major priority for primary and secondary preventive actions.

The use of the international ICD-10 classification for the encoding of pathologies that lead to a determination of unfitness would increase the quality and the processing speed of analyses. Unfortunately, the different information systems in the occupational health services do not provide such a possibility at this time. However, a common system

would considerably enhance the epidemiological capacity to analyze ill-health related job losses, and increase the processing speed.

It would be appropriate to match our findings with other data sources, such as expert-based networks or observations, as we believe that our data are complementary.

The population included in our study was limited (50,000 workers). A more significant population would increase the power of the analysis and would allow a more accurate identification of the sub groups of workers, or specific socio-professional categories, at risk of losing their jobs for medical reasons. It would also provide better information concerning prevention targets or further specific studies.

## CONCLUSIONS

French occupational health systems have an important and cost effective epidemiological capacity. The supervisory authority of the occupational health services, the General Labor Department of the Ministry of Work, should consider promoting such an epidemiological approach based on occupational health service data, which could provide useful, accurate, and reliable information in the field of occupational health. Moreover, occupational health information systems contain accurate and high quality data on workplace analysis, medical records, or sociodemographic data. Several European countries, such as Belgium, could also collect and analyze similar data (*Godderis et al., 2015*). For these very reasons, an Occupational Health System Data Analysis network should be promoted.

## ACKNOWLEDGEMENTS

The authors would like to thank the occupational physicians who participated in the study, the occupational health service AMETRA-Montpellier, and in particular Mr. Bruno Yerriah.

### Funding

The publication fees for this work were supported by the non-benefit research association IRIST (No. W 513002745). The funders had no role in study design, data collection and analysis, decision to publish, or preparation of the manuscript.

### Grant Disclosures

The following grant information was disclosed by the authors:
IRIST: W 513002745.

### Competing Interests

The authors declare that they have no competing interests.

### Author Contributions

- Francois-Xavier Lesage conceived and designed the experiments, analyzed the data, contributed reagents/materials/analysis tools, approved the final draft.

- Frederic Dutheil conceived and designed the experiments.
- Lode Godderis authored or reviewed drafts of the paper.
- Aymeric Divies conceived and designed the experiments, performed the experiments.
- Guillaume Choron performed the experiments, analyzed the data, contributed reagents/materials/analysis tools, prepared figures and/or tables, authored or reviewed drafts of the paper.

### Human Ethics

The following information was supplied relating to ethical approvals (i.e., approving body and any reference numbers):

Our study was carried out before the new French law (loi Jardé - nov 2016) was enacted. However, to conform with this new law, we received retrospective approval from our local institutional review board (2018_IRB-MTP_02-02).

### Data Availability

Researchgate: https://www.researchgate.net/publication/321071682_Job_Loss_Database.

### Supplemental Information

Supplemental information for this article can be found online at http://dx.doi.org/10.7717/peerj.5073#supplemental-information.

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
