# Peer review of "Incidence of ill-health related job loss and related social and occupational factors. The "unfit for the job" study: a one-year follow-up study of 51,132 workers"

_PeerJ, doi:10.7717/peerj.5073_

## Round 0.1 · original submission · Major Revisions

Dear authors,

I have assessed your paper with the comments of the reviewers, and it could not be accepted for publication in its current form. Therefore, you should re-send another version of the text taking into account the comments of the reviewers. Consequently, my decision is MAJOR REVISION.

With respect and warm regards,
Dr Palazón-Bru (academic editor for PeerJ)

·

Basic reporting

There are problems with syntax and terminology eg abstract, objective - medically unfit to (do) their jobs; abstract, methods - unfit workers rather than unfit patients; abstract, results - numbers as well as percentages; line 46 ...it is (an) important to ... ; line 47 - Most data about these (pathologies) - suggest - occupational diseases instead of pathologies; line 50 - (epidemiological) assessment by a specialist - I think you mean the gap between assessment by a specialist removed from the workplace and assessment by an occupational physician linked to the workplace; line 132 number unfit rather than number of unfitness; the term mental ill-health is preferable to psychopathology throughout the manuscript.

It is usual to start the discussion with the most important findings and then discuss them in a wider context. For example, there is no reference to a) the study by Whitaker and Aw in Occup Med 1995; 45: 75-80 of 9139 pre-placement assessments in the NHS in which 0.7% were found to be unfit for the job and 1.3% fit for restricted duties. Similarly most of those found to be unfit/restricted were because of obesity, MSk problems and mental ill health. b) the study by Alavinia et al in Scand J Work Environ Health 2009; 35: 325-33 of factors in long-term sickness absence in which the highest attributable fraction was those >age 50.

Experimental design

Not collecting data on 'fitness for modified work or occupational reclassification' is a weakness of the study. This figure could be much higher than those found to be unfit for their job. It is not clear to me whether the 0.778% figure relates to unfitness for job or unfitness to work for that employer or unfitness to work at all. I suggest this is made clearer.

Under Methods please define how workers were defined as blue collar.

Also line 215 refers to the global 1-year incidence of unfitness as 7.78%. Yet the proportion found to be unfit to do their job was 0.778%. I do not understand the difference.

In lines 150, 151 and 167 there are references to the proportions ill health thought to be work related by occupational physicians. How was work-related defined? Numbers should be given as well as percentages. For example, attribution of mental illness to work is thought to involve the consideration of 18 factors (11 workplace stressors and 7 personal factors of vulnerability) - see Wong et al. Occup Med 2015; 65: 391-7. This paper should be referenced to illustrate the complexity of attribution.

Validity of the findings

Under statistical analysis line 131 please explain ORc and ORa and how the adjustments were made.

·

Basic reporting

• This article was concise with a well-articulated rationale. The introduction provides a nice background of context. The literature is well-referenced and relevant.
• There are some potential English language concerns that limit clarity, e.g., “the outcome is job loss, which is an objective data…”(line 209); “this must not be considered as a limitation” (line 204); ‘the difficulties are:…’ (line 52; difficulties with what?); ‘ it is an important to set the priority actions concerning health and safety at work’ (line 46); ‘Seventeen Ops followed up employees of this employment area’ (line 108); ‘all occupation groups… were concerned’(line 171). I was also confused by the term ‘work related accountability.’ A thorough review for English language is recommended.
• The tables are extremely difficult to interpret – the variable names run together and the variable headers do not align with the information (E.g., mean(sd)) above age distributions. It is unclear whether this was a technical issue with the upload; but, I was unable to evaluate the validity of the findings because of this issue.

Experimental design

• Research design is well-defined and relevant – this manuscript addresses an important gap by examining incidence by time within a medical system vs. relying on workers’ compensation records.
• The main concern is the potential of selection bias with this region of the country in France. Because of the Table issue above, the occupation and activity sector distribution of your study is hard to assess. Is the occupation/activity sector of your sample representative of France as a whole? Does the generalizability of your findings need to be qualified by your sample? (e.g., few cases in manufacturing, construction, and information and communication). Why were farming and public service employees excluded?
• The abstract states that the design allowed ‘two data frames to be merged.’ But the Methods describe this as one case control study (line 107). Clarification the statement made in the abstract would be helpful.
• I am concerned about the recommendation that no ethics approval is required because a similar, previous study did not require ethics approval. I recommend sending this data analysis to the institutional review board to provide a letter or assurance that review is not needed.

• “The reader should be aware that this study design does not identify the work related diseases, neither can it provide a job loads attributable risk, but rather assesses the impact of diseases, work-related or not, on the capacity to maintain the current job.”
o The authors make a very good point about the focus of this manuscript on job loss vs. overall work-related disease vs. work-related accountability (As primary). The scope is well-stated.

Validity of the findings

• Please define “ORa” in the manuscript.
• See point above related to the Tables – until these numbers can be interpreted, a statistical review is difficult.
• The statement made on Line 252 about psychopathology being invisible and being ‘very difficult to determine occupational accountability’ – is probably correct but should be labelled as speculative.
• The statement made on Line 247 about workplace rarely adapted for age is also probably correct.. but should also be labelled as speculative.
• The statement made on Line 265 ‘ Traditionally, a high proportion of women work in activity sectors with high physical or psychological loads, such as health care, accommodation for seniors or social care’ requires a reference.
• Overall, the conclusions are well-stated, linking to the original research question.

Additional comments

This large-scale study provides a significant contribution to the literature in that it provides information about health-related job loss, which has not yet been explored extensively previously.

---

## Round 0.2 · Minor Revisions

Dear authors,

Before publication you should correct some minor changes suggested by the reviewers. Furthermore, it is important to clarify the relevant information regarding the French law and the IRB approval in Methods (see comments of the reviewer #2): timing for the obtention of the approval and description of the French law​.

With respect and warm regards,
Dr Palazón-Bru (academic editor for PeerJ)

·

Basic reporting

This is better than the previous draft but there are still several problems with the English, grammer, syntax, typing and meaning. As there are so many of them I will just refer to the line numbers - Abstact: methods - South of France ... analysed ... Results: occupation ... sector ... lines 61, 65, 70, 110, 139 - OP's ...the work ... 182, 190, 192 - compared with who?, 195-196, 198, 200, 217 & 221 - why is it ok to determine work-relatedness for MSk conditions and mental ill health, 231 - how can comparisons be made with other countries when we have different ways of collecting work-ralted ill health, 295, 297 - a link to how attribution was made by the French OPs needs to be made here, 299, 309, 311, 323, 334-336, 338 what is a common thesaurus, 338-342 what is meant by increasing quality and processing speed?

It is well structured with two good tables but poorly written. The main findings are well made in that older workers, blue collar workers and women are more likely to loose their jobs due to musculoskeletal disorders or mental ill health.Comparisons are made with another French study and with findings from the THOR surveillance scheme in the UK which is good.

There is no need to report unfitness for the job to two decimal places. 7.8% will do!

As the authors are suggesting more similar research with a larger population size, I should like to know how vague diagnoses like musculoskeletal disorders and mental ill-health will allow tageted preventative interventions. Surely diagnoses need to be more specific than this such as epicondylitis, tenosynovitis, mechanical back pain, anxiety, depression, PTSD, etc.

Experimental design

This is good. Multivariate analysis is appropriate.

Could the authors add how many cases were seen per day by each OP. I estimated 13. If this is the case it does not give much time for attribution to be determined, particularly when this can be complex such as for mental ill health. For example, do the OPs visit the workplace? How would a system specialist be able to evaluate the workplace factors? The large numbers of cases seen per day should be stated as a limitation with regards to attribution.

Validity of the findings

This research adds to the literature. I should like to know why it has taken 3 years to submit this work for publication.

Additional comments

The authors still need to do more proof reading. I was distracted from the main messages by the way the manuscript is written.

·

Basic reporting

The language flows better, but there is still some awkward language throughout. Also, please be sure to edit for detailed grammatical and punctuation issues. I see two periods after sentences, sentences beginning with lower case letters, etc.

Table 1 has improved but still needs a row label for the numbers included in the header (presumably these are overall sample?) Note that this is somewhat odd stylistically and these numbers in the header would be much better as a row at the bottom of the table --
since technically we still don't know that N% represents each column. Also, the incidence % needs to be clarified so that the table can stand on it's own (Incidence of what?). Table notes would be helpful.

Similarly, Table 2 includes bolding -- the significance of which is unclear without specification. The OR definitions/ adjustments are helpful on the website. It is unclear whether these are actually table notes...

"primary prevention of workplace tasks is probably rare"... does not make sense

Still no references for statement in the original Line 265 (a high proportion of women work in activity sectors with high physical or psychological loads)

Experimental design

A major concern is the conduct of research and submission of the manuscript involving personal health information without prior Institutional Review Board approval. It is unfortunate that the authors did not wait to receive this approval (or an official determination that the project is exempt from review) prior to the conduct of the study.

Validity of the findings

Overall, the authors were largely responsive but there are a few instances in which they provide rebuttal responses that are not incorporated into the manuscript.

It is recommended that the authors include the following discussion in the manuscript:
-Discussion about the limits of generalization (geographic and specific sectors not represented)
-Implications for future research (related to the extrapolation of socioprofessional categories to France in general)

Additional comments

Please note that this review was provided in concert with Rebecca Wolf, JD, MPH, OTR/L.

---

## Round 0.3 · Minor Revisions

Dear authors,

Still pending some minor modifications before acceptance of your paper. Please, address the comments of Reviewer #1.

With respect and warm regards,
Dr Palazón-Bru (academic editor for PeerJ)

·

Basic reporting

The paper is now much better written. In the section - What is new? the authors only refer to the use of mutivariate analysis. What are the new findings from this research?

Abstract - 7.8%, not 7.78%
L 63 psychiatrists
192-193 associated with rather than 'concerned by'
196 There were 153 declarations of unfitness caused by mental ill health - not ... were delivered
220 what is meant by - number of cases is underevaluated - do you mean in other surveillance schemes such as THOR?
245 subacromial impingement is not a good example here as sytematic reviews have shown this to be associated with heavy manual handling, lifting above shoulder height and throwing. Suggest mental ill health instead.
314 all workers are at risk - rather than - are concerned

Experimental design

no comment

Validity of the findings

It is still not clear to me how causality (work relatedness) for the reason for job loss was determined. I presume by the OP in the short time with the worker. How would having a larger sample size help? The large difference between causality as determined by the OP and by the insurance system is not addressed in the discussion. Why?

Additional comments

I am happy for all my comments to be shared with the authors.

·

Basic reporting

The authors have been largely responsive and addressed major concerns.

Experimental design

The authors have been largely responsive and addressed major concerns.
In the text, please state the name of the IRB that provided approval.

Validity of the findings

The authors have been largely responsive and addressed major concerns.

---

## Round 0.4 · accepted · Accept

Dear authors,

I have reviewed the previous comments of the reviewer and your own responses, and I consider that your paper has high standards to be published in PeerJ in its current form,

Congratulations!

With respect and warm regards,
Dr Palazón-Bru (academic editor for PeerJ)